# A Cross-Sectional Study of Grocery Shopping Factors of Importance among Food-Insecure African Americans

**DOI:** 10.3390/nu16081188

**Published:** 2024-04-17

**Authors:** Cedric Harville II, Delores C. S. James, Amaria Patterson, Sheila Harper, Lindy Petchulat-McMillan

**Affiliations:** 1Department of Applied Health, Southern Illinois University-Edwardsville, Campus Box 1147, Edwardsville, IL 62026, USA; amapatt@siue.edu (A.P.); sharper@siue.edu (S.H.); limcmil@siue.edu (L.P.-M.); 2Health Education & Behavior, University of Florida, 1864 Stadium Road, Gainesville, FL 32603, USA; djames@hhp.ufl.edu

**Keywords:** African Americans, food insecurity, grocery shopping, obesity, SNAP BMI, nutrition, health disparities, government assistance

## Abstract

The objective of this study was to (1) assess via cross-sectional survey the prevalence of food insecurity among African Americans [AAs] after their most recent grocery shopping trip, and (2) examine the grocery shopping factors of importance and characteristics of food-insecure AA grocery shoppers. Most (70.4%) were food-insecure. Food-insecure grocery shoppers were significantly more likely to be younger, less educated, who often skipped meals and/or practiced fasting, accessed a food pantry, were SNAP recipients, were considered to not be in ”good” health, and who had higher BMI compared to food-secure shoppers (*p* ≤ 0.03 * for all). Our data showed that AAs shopped for groceries a mean 2.20 ± 1.29 times per week, for low prices (72.1%), without a weekly budget (58.9%), with a grocery list (44.6%) or using an app (27.6%), for high-quality vegetables (27.5%), for good customer service (22.9%), for store brands (20.8%) and name brands (17.9%).Food-insecure shoppers were significantly more likely to grocery shop more times per week, have a weekly budget, and use an app, but were significantly less likely to report store brands, name brands, good customer service, and high-quality vegetables as grocery factors of importance (*p* ≤ 0.03 * for all). Grocery strategies such as shopping with a grocery app and/or grocery list could help food-insecure AAs reduce grocery trips, promote meal planning to save money, and avoid skipping meals/fasting, while eating healthier.

## 1. Introduction

Food insecurity refers to limited “access by all people at all times to have enough food for an active, healthy life” [1]. Nationally, the rate of food insecurity is around 12.8% [1]. The current rate of food insecurity is the highest on record since 2015 (12.7%) and the beginning of the COVID-19 pandemic (10.5%) [1]. Unfortunately, disparities in food insecurity compared to the national average can be identified among African Americans. African Americans have been found to have a significantly higher rate of food insecurity compared to the national average (22.4% vs. 12.8%) and White Americans (22.4% vs. 9.3%) [1]. The overall rate for African Americans has increased from 19.8% in 2021 to the current rate of 22.4% [1].

A recent review found many factors contributing to food insecurity among African Americans inclusive of low socioeconomic status, employment status, household income, gender, single-parent home, obtaining government assistance, managing chronic conditions, reliance on others for transportation and income supplementation, availability of grocery stores, and fresh produce, amongst others [2]. Others have explored the impact of racial discrimination as a factor related to disproportionate food insecurity among African Americans due in part to a lack of employment as a result of being less educated, where African Americans are more likely to experience racial discrimination in the workforce [3]. The Supplemental Nutrition Assistance Program (SNAP) was developed to assist individuals and families with limited income to obtain food [4]. Studies have suggested that SNAP has had a positive effect in reducing the risk of food insecurity among African Americans in comparison to Whites [5]. A majority of the African American community, where SNAP benefits may be used for food purchases, might have limited access to nutritious-quality food options that could possibly further increase food costs [6]. Lastly, the impact of SNAP benefits due to rising food costs may have been further reduced for African Americans due to the COVID-19 pandemic and associated inflation [7,8]. Despite this finding, a majority of African American-concentrated communities might not be able to effectively utilize SNAP due to factors such as redlining, which has led to reduced food access and quality grocery stores [9].

### Literature Review

Research has shown that African American communities have the fewest number of grocery stores compared to all other races and levels of poverty [10]. Reduced food access results in limited access to grocery stores, fresh fruits and vegetables, and to living in areas more populated with convenience stores and fast-food establishments [11]. Similarly, studies have shown an association between residents who live in food deserts that frequently visit grocery stores with increased BMI [12], which contributes to an increased risk for preventable chronic diseases and, more recently, COVID-19 [13,14,15]. One study followed the model of grocery stores using meal kits to help provide healthy food access to African Americans; however, the cost remains a barrier [16]. Another concern among grocery shoppers in major urban areas is the lack of quality foods and having to travel distances outside the neighborhood for better and cheaper foods [17]. Interestingly, African Americans have been found to less likely travel further distances for groceries and visit the grocery store less often compared to Hispanics and Whites [18,19]. Increased trips to the grocery store have been associated with health-promoting behaviors such as increased fruit and vegetable consumption [20]. Even with the increased availability of online grocery shopping, low-income grocery shoppers have been found to be less likely to take advantage of these services due to fees and the inability to manage the selection of fresh foods and produce [21,22]. All of the above could potentially lead to poor diet quality and increased food insecurity experienced among African Americans [23].

Cost and distance from quality food options have been discussed as common issues related to grocery shopping and access. However, further exploration of grocery shopping factors of importance could add more substantive data regarding what food-insecure African Americans value when grocery shopping. *The goal of this study was (1) to determine the prevalence of food insecurity among African Americans after their most recent grocery shopping trip, and (2) to examine the grocery shopping factors of importance and the characteristics of food-insecure African American grocery shoppers.*

## 2. Materials and Methods

### 2.1. Participant Recruitment

African Americans were recruited within the East St. Louis, Illinois, community between the dates of 2 September 2023 and 27 September 2023. East St. Louis, Illinois, is a majority African American community (97%) located on the border of Missouri and Illinois. East St. Louis, Illinois, is one of the poorest communities within the state of Illinois as the median income for these residents is over 2.75 times lower than that of Illinois residents (USD 24,009 vs. USD 68,428) and US residents (USD 24,009 vs. USD 64,994) [24].

To participate in this study, individuals had to be 18 years and/or older, have the ability to read and write in English, and self-identify as African American. Potential participants were intercepted upon exiting the only local major branded grocery store within East St. Louis, Illinois, by the research team to take a written cross-sectional survey. Prior to the beginning of data collection, the PI received approval from the store manager, regional manager, and district manager, to set up a table directly in front of the grocery store, and approach potential study participants once they completed their individual grocery shopping trips and exited the grocery store. Upon exiting the grocery store, potential participants were approached by either the PI and/or research assistant(s) to gauge interest in taking a 10–15 min survey related to their grocery shopping habits and physical health. Potential participants were provided a USD 20 gift card to the grocery store they just exited as compensation for their time.

### 2.2. Instrument

The self-administered survey consisted of sociodemographic characteristics (24 items), measure of food insecurity (8 items), grocery shopping items (5 items), and resources to make better food choices (1 item). The survey was developed using the Qualtrics survey platform. Upon completion, the survey was exported to Microsoft Word and printed for use during data collection. 

### 2.3. Data Analysis

#### 2.3.1. Sociodemographic Characteristics

Sociodemographic variables included the following: race (African American/other), gender (male/female), parent of a child 18 years or younger, total number of people in the home, highest level of education, current employment status (employed/unemployed), hours worked per week, annual household income, homeownership, SNAP status (yes/no and SNAP dollars spent), accessed food pantry (yes/no), skipped meals (yes/no), fasted (yes/no), chronic disease status (hypertension, diabetes, heart disease, cancer, stroke), shop with grocery list (yes/no), weekly grocery shopping budget (yes/no), shop with grocery app (last 30 days), and number of weekly grocery shopping trips. Participants reported their individual health rating by self-report (excellent, very good, good, fair, or poor). For analysis, health rating was dichotomized to (1) at least “good” (excellent, very good, good) and (2) “not good” (fair/poor). Participants self-reported height, current weight, and weight gained this year. Chi-squared analyses were performed among the categorical sociodemographic variables and food security status (food-insecure vs. food-secure shoppers). Height (feet and inches) and weight (pounds) were self-reported. Height was reported in feet (ft) and inches and was converted to inches. BMI was calculated within the dataset using the equation 703 × (lbs/[in^2^]). Differences in mean BMI, weight, weight gained this year, SNAP dollars spent, food security status, and sociodemographic variables (continuous) were examined using one-way analysis of variance (ANOVA). Significance was established at *p* < 0.05 level for all statistical tests. Mean BMI, weight, weight gained this year, and SNAP dollars spent were reported with standard deviations [SD] (i.e., BMI ± SD; weight ± SD; SNAP ± SD).

A final dataset was created by combining the two exported Excel spreadsheets. A total of 485 individuals participated in this study. After the data were cleaned to account for incomplete responses (n = 5), the final dataset is reflected as n = 480. Data were analyzed using JMP Pro 17.1 software [25].

#### 2.3.2. Measure of Food Insecurity

The 6-item Short-Form USDA Household Food Security Survey Module was used to measure food insecurity [26]. The scale has been previously validated in research among African Americans [20,27]. Participants were scored based on the number of positive responses to the six items on the scale. Including the reported response of “yes”, positive responses that were considered consisted of “sometimes” and “often”. Those who reported positive responses that led to a cumulative raw score of 2 or greater were considered “food-insecure”. Participants with any cumulative raw score that was less than 2 were considered “food-secure”. This scoring was consistent with previous studies in the literature that categorized food security status into two different groups, “food-insecure” and “food-secure”, among African Americans [27]. A total of 480 individuals participated in the study and completed the USDA Food Security Scale in total. Internal consistency for the 6-item Short-Form USDA Household Food Security Survey Module was α = 0.79. There are six-items from the USDA Food Security—Short Form, which measure food insecurity experienced within the last 12 months. These six items are as follows. “In the last 12 months, the food that (I/we) bought just didn’t last, and (I/we) didn’t have money to get more”. “In the last 12 months, (I/we) couldn’t afford to eat balanced meals”. “In the 12 months, since last (name of current month), did (you/you or other adults in your household) ever cut the size of your meals or skip meals because there wasn’t enough money for food?” “How often did this (cut the size of or skip meals) happen—almost every month, some months but not every month, or in only 1 or 2 months?” “In the last 12 months, did you ever eat less than you felt you should because there wasn’t enough money for food?” “In the last 12 months, were you ever hungry but didn’t eat because there wasn’t enough money for food?” [26].

#### 2.3.3. Measure of Grocery Shopping Factors

The following item measured grocery shopping factors: “What are the most important factors that you consider when you buy groceries?” There were 11 different categorical responses: low prices, good/wide selection, high-quality fruits, high-quality vegetables, food on sale, organic options, close to my home, needs of other family members, good customer service, name brands, and store brands. Participants were asked to “choose all that apply” among the listed factors of importance when grocery shopping.

#### 2.3.4. Measure of Resources to Make Better Food Choices

The following item measured resources to make better food choices: “What type of resources do you need to make better food choices at the grocery store?” These resources included access to a nutritionist, quick healthy recipes, a commercial diet program, cooking classes, learning to eat on limited budget, applying for SNAP, and where to find food banks. Participants were asked to “choose all that apply” among the listed resources necessary to make better food choices.

## 3. Results

### 3.1. Food Insecurity

All (100%, n = 480) participants reported their race as African American. Most participants were food-insecure (70.4%, n = 338). Responses to the 6-item Short-Form USDA Household Food Security Survey Module can be found in Table 1. A total of 44.6% (n = 214) were considered to have “low food security”, and 25.8% (n = 124) were considered to have “very low food security”. Most participants were single (84.6%, n = 405), had a high school education or lower (70.7%, n = 338), were female (55.7%, n = 267), and parents of children ages 18 or younger (51.0%, n = 238). Those who were food-insecure were significantly more likely to have a high school education or lower [74.8% vs. 61.4%; *X*^2^(1) = 8.54; OR = 0.54; 95%(CI) = 0.35–0.82; *p* < 0.01 *] compared to those who were not. The mean age of participants was 52.84 ± 14.32 years. Those who were food-insecure were significantly younger in age compared to those who were food-secure (*t*_449_ = 6.05; 51.75 ± 14.09 vs. 55.35 ± 14.58; *p* = 0.01 *). Most participants were unemployed (61.1%, n = 291). The remaining participants who were employed (38.9%, n = 185) worked a mean of 34.50 ± 10.63 h per week. The mean annual income of participants was USD 21,841.27 ± USD 17,657.35. Those who were food-insecure earned significantly less annual income compared to those who were food-secure (*t*_331_ = 17.49; USD 19,225.93 ± USD 15,096.91 vs. USD 27,822.89 ± USD 21,350.70; *p* < 0.0001 *). The average number of people that live in each household was 2.67 ± 1.78 people. Most were not homeowners (73.2%, n = 341).

### 3.2. Health Status

Most participants rated their health as at least “good” (66.2%, n = 313). Those who were food-insecure were significantly less likely to report their health as “good” compared to those who were food-secure [63.2% vs. 73.4%; *X*^2^(1) = 4.68; OR = 0.62; 95%(CI) = 0.40–0.96; *p* = 0.03 *].

#### 3.2.1. Weight Status

The mean weight of participants was 185.58 ± 46.11 pounds. Those who were food-insecure weighed significantly more compared to those who were food-secure (189.72 ± 44.87 vs. 176.01 ± 47.37 pounds; *t*_452_ = 8.59; *p* < 0.01 *). Nearly 42.2% of participants reported that they felt like they gained weight since the start of this year. Those who were food-insecure were significantly more likely to report that they gained weight since the start of this year compared to those who were food-secure [45.7% vs. 33.8%; *X*^2^(1) = 4.68; OR = 0.60; 95%(CI) = 0.40–0.92; *p* = 0.02 *]. Those who were food-insecure gained more weight since the beginning of the year compared to those who were food-secure (17.59 ± 29.94 vs. 10.54 ± 8.07 pounds). However, no significant differences were found in weight gained this year (*p* > 0.05). The mean BMI for participants was 29.51 ± 7.68. Significant differences in BMI were found based on food security status. Those who were food-insecure had a significantly higher mean BMI compared to those who were food-secure (30.27 ± 7.90 vs. 27.84 ± 6.91; *t*_441_ = 9.66; *p* < 0.01 *). Significant differences in BMI were found based on food security status and gender. Female food-insecure participants had a significantly higher mean BMI compared to male food-insecure participants (31.82 ± 8.61 vs. 28.39 ± 6.49; *t*_303_ = 14.84; *p* = 0.0001 *).

Most participants (78.3%, n = 376) reported that they were currently trying to lose weight. Participants who were food-insecure were significantly more likely to be currently trying to lose weight compared to those who were food-secure [82.5% vs. 68.3%; *X*^2^(1) = 11.94; OR = 0.46; 95%(CI) = 0.29–0.72; *p* < 0.001]. Most reported that they did not skip meals (87.9%, n = 422) or fast (87.1%) to lose weight. Those who were food-insecure were significantly more likely to skip meals [14.8% vs. 5.6%; *X*^2^(1) = 8.97; OR = 2.91; 95%(CI) = 1.34–6.31; *p* < 0.01] and fast [15.7% vs. 6.3%; *X*^2^(1) = 8.73; OR = 2.75; 95%(CI) = 1.32–5.74; *p* < 0.01] compared to those who were food-secure.

Participants were asked to rate their individual health. Most participants (70.9%, n = 336) reported their individual health as “good”. Those who were food-insecure were significantly less likely to report their individual health as “good” compared to those who were food-secure (66.0% vs. 82.7%; *X*^2^(1) = 14.28; OR = 2.47; 95%(CI) = 1.51–4.05; *p* < 0.001).

#### 3.2.2. Chronic Disease Status

Participants were asked to report if they were currently managing or had been diagnosed with a number of diseases and/or conditions. No majority was reported. However, participants had been diagnosed with hypertension (32.1%, n = 154), diabetes (13.1%, n = 63), heart disease (6.0%, n = 29), cancer (5.8%, n = 28), and stroke (3.5%, n = 17). Those who were food-insecure were significantly more likely to have been diagnosed with heart disease compared to those who were food-secure [7.2% vs. 1.7%; *X*^2^(1) = 6.07; OR = 4.49; 95%(CI) = 1.04–19.30; *p* = 0.01 *]. Food-insecure shoppers were more likely to be hypertensive (33.4% vs. 28.9%) but less likely to have diabetes (12.1% vs. 15.5%), cancer (5.6% vs. 6.3%), or stroke (3.0% vs. 4.9%) [*p* > 0.05 for all].

### 3.3. Grocery Shopping

Most participants were SNAP recipients (65.0%, n = 308). Those who were food-insecure were significantly more likely to be SNAP recipients compared to those who were food-secure [69.2% vs. 55.0%; *X*^2^(1) = 8.69; OR = 1.83; 95%(CI) = 1.22–2.75; *p* < 0.01 *]. Participants spent in SNAP on average USD 59.88 ± USD 64.21 at the grocery store. Those who were food-insecure spent more in SNAP compared to those who were food-secure (USD 70.73 ± USD 78.77 vs. USD 49.02 ± USD 49.65; *p* > 0.05 *). Most participants (65.8%, n = 313) reported they accessed a food pantry. Those who were food-insecure were significantly more likely to have access to a food pantry compared to those who were food-secure [70.5% vs. 54.6%; *X*^2^(1) = 10.82, OR = 1.98; 95%(CI) = 1.32–2.97; *p* < 0.001 *]. Participants reported that they normally grocery shop with a grocery list (44.6%, n = 199). Those who were food-insecure were significantly more likely than those who were food-secure to report normally shopping with a grocery list [48.7% vs. 35.0%; *X*^2^(1) = 7.53; OR = 1.76; 95%(CI) = 1.17–2.65; *p* < 0.01 *]. Participants reported using a grocery shopping app within the past 30 days to help guide their shopping (27.6%, n = 131). Those who were food-insecure were significantly more likely to use a grocery shopping app within the last 30 days to guide their shopping compared to those who were food-secure [31.9% vs. 17.3%; *X*^2^(1) = 11.11; OR = 0.45; 95%(CI) = 0.27–0.73; *p* < 0.001 *]. Most participants reported not having a weekly grocery shopping budget (58.9%, n = 264). Those who were food-insecure were significantly more likely to have a grocery shopping budget compared to those who were food-secure [44.9% vs. 32.1%; *X*^2^(1) = 6.76; OR = 1.72; 95%(CI) = 1.13–2.61; *p* < 0.01 *]. Most participants (87.3%, n = 405) reported that they most often shopped for groceries at “this grocery store”. Food-insecure participants were significantly more likely to grocery shop most often at “this grocery store” compared to those who were food-secure [89.3% vs. 82.6%; *X*^2^(1) = 3.68; OR = 1.75; 95%(CI) = 1.00–3.07; *p* = 0.05 *]. Participants reported that they went to the grocery store a mean 2.20 ± 1.29 times per week. Food-insecure participants went to the grocery store significantly more times per week compared to those who were food-secure [2.63 ± 1.98 vs. 2.16 ± 1.16 times per week; *t*_378_ = 5.46; *p* = 0.02 *].

#### 3.3.1. Factors of Importance While Grocery Shopping

Participants reported on the most important factors they consider when purchasing groceries from the grocery store. These factors include low prices, good/wide selection, high-quality fruits, high-quality vegetables, food on sale, organic options, grocery store close to home, needs of other family members, good customer service, name brands, and store brands. Most participants reported low prices (72.1%, n = 346) as the most important grocery shopping factor. Other responses include the following: high-quality fruits (33.1%, n = 159), good/wide selection (30.0%, n = 144), high-quality vegetables (27.5%, n = 132), grocery store close to home (26.3%, n = 126), good customer service (22.9%, n = 110), store brands (20.8%, n = 100), name brands (17.9%, n = 86), food on sale (16.5%, n = 79), needs of other family members (9.6%, n = 46), and organic options (7.1%, n = 34). Food-insecure participants were found to be significantly less likely to report store brands, name brands, good customer service, and high-quality vegetables as important factors when purchasing groceries compared to food-secure participants (*p* < 0.05 * for all). See Table 2 for the distributions.

#### 3.3.2. Resources to Make Better Food Choices While Grocery Shopping

Participants reported on the resources to make better food choices at the grocery store. These resources to make better food choices include: learning to eat on a limited budget (40.2%, n = 193), quick healthy recipes (39.0%, n = 187), access to a nutritionist (21.0%, n = 101), cooking classes (20.2%, n = 97), a commercial diet improvement program (15.0%, n = 72), where to find food banks (14.6%, n = 70), and applying for SNAP (9.7%, n = 47). Food-insecure participants were significantly more likely to report a commercial diet improvement program as a resource to make better food choices at the grocery store (*p* = 0.02 *). See Table 3 for the distributions.

## 4. Discussion

In the current study, 70.4% of participants were found to be food-insecure, which is nearly six times higher compared to the national average of household food insecurity at 12.8% [1]. Participants in the current study were food-insecure at a rate greater than three times the national average for African Americans [70.4% vs. 22.4%], who experience the highest levels of food insecurity compared to all other racial and ethnic groups [1]. Such disproportionate food insecurity numbers are especially concerning especially as data were collected from participants upon completion of their grocery shopping trip. Specifically, among racial and ethnic groups such as African Americans, food insecurity has been linked to factors such as low socioeconomic status, low employment, and environmental factors such as limited access to social resources and food stores [28]. Along the same lines, we found in the current study that participants were found to be significantly more likely to have completed lower than a high school education. Interestingly, one recent study of African Americans did not find a significant difference based on education level for food insecurity [29]. This might be due to a number of factors such as sample size and the use of a different scale to measure food insecurity [29]. Consequently, a majority of African American neighborhoods are more likely to be living in food deserts which possibly limits their availability of grocery stores and access to healthy food options [30]. Interestingly, less than five months before data were collected for this study, one major food store and another nationally branded supermarket abruptly closed in the East St. Louis, Illinois, community [31]. Factors such as the abrupt closing of grocery stores might negatively contribute to the food insecurity burden African Americans currently experience locally and nationwide [10,11]. The annual mean household income for food-insecure participants was found to be significantly lower compared to food-secure participants. Food-insecure annual mean household income was over USD 2000 less than the sample mean income, over USD 8000 less than food-secure shoppers, and nearly USD 5000 less compared to the reported mean income from the US census for those who were food-insecure East St. Louis residents [24]. SNAP was created to help low-income and food-insecure families in poverty to escape hunger [4]. Participants in this study accessed SNAP at higher levels compared to the national average of SNAP recipients [65.0% vs. 55%] [1]. Also, in the current study, food-insecure participants were significantly more likely to be SNAP recipients (69.2% vs. 55.0%) compared to those who were food-secure. Interestingly, we found that food-insecure individuals spent on average USD 21 more at the grocery store compared to food-secure individuals. SNAP appears to be providing more benefits to those who might be food-insecure, as how it was designed. However, those same participants in the current study might be struggling with food insecurity due to a few factors. First, as a result of the COVID-19 pandemic, SNAP recipients received Emergency Allotments (EAs) of benefits as part of the Coronavirus Aid, Relief, and Economic Security (CARES) Act, which on average increased SNAP by USD 95 nationwide [Illinois—USD 171/household; USD 86/person] [32,33,34]. Currently, grocery shopping receipt data found that participants spent on average nearly USD 60 in SNAP benefits. For Illinois residents, with the CARES EA plus SNAP money spent at the grocery store would have seen individuals receiving nearly USD 146 and families receiving USD 231 from the state for groceries six months prior to data collection. These funds expired in totality across the US and more specifically Illinois in March 2023 [33]. Second, inflation of food prices made food less affordable for all Americans [7]. The price of groceries increased nearly 10% in 2022, that had an expected increase of nearly 6% for 2023 [7]. It is possible the loss of funds from the CARES EA coupled with the sharp inflation of foods could have made a significant impact in why we saw such a high rate of food insecurity in the current sample.

Research has shown that African Americans who reside in food deserts are more likely to be less educated and are more likely to live within poverty-stricken areas [35]. Consequently, those living within food deserts are more likely to select unhealthy foods for meals [23]. Thus, African Americans living in food deserts are possibly at a greater risk for obesity [36]. Recent reviews have found food insecurity associated with obesity [37]. Similarly, in the current study, food-insecure African Americans were significantly more likely to be obese. One study explored overweight and obesity among adults and found that food-insecure African Americans had significantly higher mean BMI compared to White and Hispanic Americans [38]. However, similar to the current study, a breakdown of African American BMI by gender found that food-insecure African American females had a mean BMI in the obese category [38].

The known trends regarding the impact of obesity on health and chronic disease diagnosis cannot be overstated [39]. Recent national data has shown that 83.2% of Americans rate their health “good or better” however, the rate for African Americans within the state of Illinois is around 78.8%, and is even lower within the current study sample (70.9%) [40]. Recent national data have shown that the rate of hypertension is at 32.2%; however, the rate for African Americans within the state of Illinois is around 42%. Both rates are higher compared to the rate of hypertension within this study sample [41]. Along the same lines, studies have shown that African Americans who grocery shop to reduce their risk for chronic disease might struggle to access healthy food options while living within a food desert [23]. Also, studies have suggested that those who grocery shop in urban areas are more likely to purchase packaged foods with the highest calories per day compared to all other types of food stores [42].

Previous research has shown that African Americans might spend less money on groceries; however, this might mean that these shoppers are purchasing more economical foods, lower-quality foods, and might spend more money per item if they reside within a food desert [43]. Also, we found that food-insecure shoppers were significantly more likely compared to food-secure ones to be skipping meals and fasting. Skipping meals and fasting might be a result of not having enough money to spend on food among food-insecure shoppers as well. Interestingly, we found that food-insecure individuals were significantly less likely to purchase store-branded foods compared to food-secure individuals. Generally, store-branded food items are cheaper in comparison to the name brands and have been found to save shoppers up to 40% on foods [44]. Purchasing these foods might allow for food-insecure shoppers to stretch their food budgets in order to purchase more items per grocery shopping trip. It is possible that food-insecure shoppers might not see store-branded foods as having the same or similar quality from the grocery store available in their community compared to the name brands.

Food-insecure shopping priorities might point to the importance of cost when grocery shopping and a limited access to healthy vegetables for consumption [45]. Studies have shown that cost might not be as much of a barrier as a vegan diet can reduce grocery shopping bills over USD 500 per year [46]. Research has shown that adherence to a vegan diet could possibly improve the health outcomes among African Americans [47]. However, this might be a struggle for those who are food-insecure due to studies finding that such individuals have lower self-efficacy with regard to eating healthier and planning meals that may include vegetables [48]. The concept of self-efficacy within food-insecure individuals (although not measured) in the current study might potentially account for why these participants were more likely to visit the grocery store more times per week compared food-secure shoppers. From the current study, it appears that food-insecure participants are open to becoming healthier by wanting to change their diet and lose weight. Food-insecure participants were significantly more likely to want to participate in commercial diet programs compared to food-secure participants. With the sample having a mean age of 51 years, providing access to weight loss programs could be key to improving health outcomes among African Americans [49,50]. More specifically, recent interventions have shown to be effective in weight loss for food-insecure individuals enrolled in obesity treatment programs [51]. In the current study, food-insecure participants were significantly more likely to use a grocery shopping app to guide their grocery shopping within the past 30 days. Also, food-insecure participants were significantly more likely to report having a weekly grocery shopping budget. The use of smartphones by food-insecure African Americans might point to the importance of planning grocery shopping trips prior to visiting the store in order to account for cost and limit excess spending. Also, the use of technology and shopping apps could help food-insecure individuals develop grocery lists and meal plans, which could also promote healthy eating. Our study has shown that food-insecure African Americans significantly want more resources to help make quick healthy recipes and possible access to a commercial diet program for weight loss. Research has shown that the inclusion of smartphones and other technology might be beneficial in promoting African American participation in weight management programs [52,53]. Having access to a nutritionist, which food-insecure African Americans were more receptive to compared to food-secure AAs in this study, might also help food-insecure African Americans while grocery shopping to meet their individual goals of eating healthier. We also found that organic options and needs of the family were important grocery shopping factors for food-insecure participants compared to food-secure. Understanding the impact of how low-income food shoppers evaluate food cost to address resources and needs can help to provide insight into the barriers and motivators to eating healthier among food-insecure African Americans [54].

## 5. Conclusions

The goal of this study was to determine the prevalence of food insecurity among African Americans after their most recent grocery shopping trip and examine the grocery shopping factors of importance and characteristics of food-insecure African American grocery shoppers. It was found that food insecurity for African Americans surpassed the national average nearly sixfold. Even when a majority of African Americans were on SNAP and were on average provided more SNAP benefits, we found food insecurity at a disproportionate rate. This might suggest that despite food-insecure individuals averaging higher mean SNAP benefits, a shortfall still exists for these shoppers, which forces them to skip meals, fast, and effectively manage hunger. Also, food-insecure participants were significantly younger, less educated, and earned significantly lower annual income compared to food-secure ones. This finding might suggest that food-insecure individuals are most likely unemployed or underemployed and are therefore possibly struggling to become more food-secure over a prolonged period of time. Also, those who were food-insecure were found to have higher mean BMI, weight, and gained more weight in 2023 compared to food-secure participants. Along the same lines, food-insecure shoppers in this study reported by percentage that they wanted access to a nutritionist, help with making quick healthy recipes, and a commercial diet program. However, food-insecure participants were found to less likely be in “good” health. This suggests that food insecurity itself might lead to eating foods that are available regardless of nutritional value. However, these findings might also explain why food-insecure shoppers were significantly more likely to want a commercial diet program. A commercial diet program or nutritionist/dietitian might provide the necessary structure to help food-insecure individuals eat what they want (healthy or not) and manage their grocery budgets, while still maintaining or attempting to lose weight.

Opportunities exist to encourage food-insecure shoppers to grocery shop in better ways to maintain and improve their health. Future studies should engage with food-insecure individuals by focusing on healthier grocery shopping styles for better overall health of the entire family [adult(s) and children]. More specifically, the inclusion of strategies (using grocery list, grocery shopping app, etc.) that might adequately assist with bulk purchases to (1) reduce the number of trips to the grocery store and (2) promote meal planning to possibly save money, while avoiding skipping meals and fasting, and simultaneously eating healthier is highly recommended.

## Figures and Tables

**Table 1 nutrients-16-01188-t001:** African American participant responses to the USDA Food Security—Short Form (n = 480).

Food Security Items	n	%
“The food that I bought just didn’t last, and I didn’t have money to get more”		
Often true	140	29.2
Sometimes true	208	43.3
Never true	76	15.8
Don’t know	56	11.7
“In the last 12 months, I couldn’t afford to eat balanced meals”		
Often true	109	22.7
Sometimes true	211	43.9
Never true	117	24.6
Don’t know	42	8.8
In the last 12 months, since last (name of current month), did you ever cut the size of your meals or skip meals because there wasn’t enough money for food?		
Yes	195	40.7
No	237	49.5
Don’t know	47	9.8
How often did this happen—almost every month, some months but not every month, or in only 1 or 2 months?		
Almost every month	72	36.7
Some months but not every month	69	35.2
Only 1 or 2 months	42	21.4
Don’t know	13	6.6
In the last 12 months, did you ever eat less than you felt you should because there wasn’t enough money for food?		
Yes	197	41.6
No	225	47.5
Don’t know	52	11.0
In the last 12 months, were you ever hungry but didn’t eat because there wasn’t enough money for food?		
Yes	162	34.0
No	276	58.0
Don’t know	38	8.0

**Table 2 nutrients-16-01188-t002:** Chi-squared analysis of factors of importance while grocery shopping by food security status.

Grocery Shopping Factors	Yes (%)	No (%)	OR (95%CI)	*X* ^2^	*p*
Low prices					
Food-secure	101 (71.1)	41 (28.9)	1.06 (0.69–1.65)	0.09	0.76
Food-insecure	245 (72.5)	93 (27.5)			
Good/wide selection					
Food-secure	46 (32.4)	96 (67.6)	0.85 (0.56–1.30)	0.55	0.46
Food-insecure	98 (29.0)	240 (71.0)			
High-quality fruits					
Food-secure	55 (38.7)	87 (61.3)	0.70 (0.47–1.06)	2.82	0.09
Food-insecure	104 (30.8)	234 (69.2)			
High-quality vegetables					
Food-secure	52 (36.6)	90 (63.4)	0.54 (0.35–0.82)	8.16	<0.01 *
Food-insecure	80 (23.7)	258 (76.3)			
Foods on sale					
Food-secure	28 (19.7)	114 (80.3)	0.72 (0.43–1.20)	1.52	0.22
Food-insecure	51 (15.1)	287 (84.9)			
Organic options					
Food-secure	9 (6.3)	133 (93.6)	1.18 (0.54–2.60)	0.17	0.67
Food-insecure	25 (7.4)	313 (92.6)			
Store close to home					
Food-secure	40 (28.2)	102 (71.8)	0.87 (0.56–1.35)	0.38	0.54
Food-insecure	86 (25.4)	252 (74.6)			
Needs of family members					
Food-secure	12 (8.5)	130 (91.5)	1.21 (0.61–2.41)	0.31	0.58
Food-insecure	18 (10.1)	378 (89.9)			
Good customer service					
Food-secure	46 (32.4)	96 (67.6)	0.49 (0.31–0.76)	10.25	<0.01 *
Food-insecure	64 (18.9)	274 (81.1)			
Name brands					
Food-secure	34 (23.9)	108 (76.1)	0.58 (0.36–0.94)	4.98	0.03 *
Food-insecure	18 (15.4)	286 (84.6)			
Store brands					
Food-secure	41 (28.9)	101 (71.1)	0.52 (0.33–0.82)	7.90	<0.01 *
Food-insecure	59 (17.5)	279 (82.5)			

* *p* < 0.05.

**Table 3 nutrients-16-01188-t003:** Chi-squared analysis of resources to make better food choices by food security status.

Resources	Yes (%)	No (%)	OR (95%CI)	*X* ^2^	*p*
Nutritionist					
Food-secure	24 (16.9)	118 (83.1)	1.45 (0.87–2.41)	2.14	0.14
Food-insecure	77 (22.8)	261 (77.2)			
Quick healthy recipes					
Food-secure	47 (33.1)	95 (66.9)	1.42 (0.95–2.16)	2.95	0.09
Food-insecure	140 (41.4)	198 (58.9)			
Commercial diet program					
Food-secure	13 (9.1)	129 (90.9)	2.10 (1.11–3.96)	5.85	0.02 *
Food-insecure	59 (17.5)	279 (82.5)			
Cooking classes					
Food-secure	22 (15.5)	120 (84.5)	1.56 (0.92–2.62)	2.89	0.09
Food-insecure	75 (22.2)	263 (77.8)			
Apply for SNAP					
Food-secure	9 (6.3)	133 (93.7)	1.87 (0.88–3.98)	2.94	0.22
Food insecure	38 (11.2)	300 (88.8)			
Where to find food banks					
Food-secure	27 (19.0)	115 (81.0)	0.62 (0.37–1.05)	3.05	0.09
Food-insecure	43 (12.7)	295 (87.3)			
Eating on limited budget					
Food-secure	58 (40.9)	101 (59.1)	0.96 (0.65–1.44)	0.03	0.85
Food-insecure	135 (39.9)	203 (60.1)			

* *p* < 0.05.

## Data Availability

The data presented in this study are available on request from the corresponding author. The data are not publicly available due to FERPA regulations.

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
