# Peer review of "A Cross-Sectional Study of Grocery Shopping Factors of Importance among Food-Insecure African Americans"

_nutrients, 2024, doi:10.3390/nu16081188_

Round 1

Reviewer 1 Report

Comments and Suggestions for Authors

The article shows some imbalances between the chapters, which should be corrected to balance the results obtained. Nevertheless, if the methodological weaknesses are overcome, it presents interesting conclusions on the subject.

I suggest the following changes/corrections:

Abstract: should be rewritten according to the characteristics that should be present in an abstract. Too much methodological information is presented here, which makes no sense. The main results and conclusions should be presented.

Materials and methods: This is the article's main weakness. 

The sample must be characterized, and the classification variables used must be clearly presented.

It should be explained how it was constructed and whether it is representative. If not (as the article indicates), the title should not pretend to be a study of the "African Americas". There must be methodological clarity and it must be clear whether this is an exploratory study.

Conclusions: the discussion of the results is extensive and should allow for deeper and more robust conclusions in addition to answering the two objectives previously formulated.

Reviewer 2 Report

Comments and Suggestions for Authors

The topic covered by the authors is interesting. Research objectives have been clearly indicated in the article. However, the article contains some errors and shortcomings that should be taken into account before its possible publication.

1. Please think again about the title of the article - in this form it has no scientific connotation.

2. The abstract is too detailed - it contains too many research results. Only the most important ones should be included.

3. You cannot base an entire paragraph on just one reference - giving it several more times - practically after every sentence.

4. There is no indication of the originality and novelty of the research conducted.

5. The article does not include a "Literature review" section that would allow for the creation of a theoretical framework for the research.

6. In the "Materials and methods" section, there is no coherent description of the survey questionnaire used in the research - how many questions did it consist of? What type of questions were these? What were they about? Were the responses scaled?

7. The article should be refined in terms of style, punctuation and editing.

8. Of the 51 references, 10 are not very recent - they were published 10 or more years ago. Please make appropriate corrections.
